# Molecular Signaling Pathways and MicroRNAs in Bone Remodeling: A Narrative Review

**DOI:** 10.3390/diseases12100252

**Published:** 2024-10-12

**Authors:** Monica Singh, Puneetpal Singh, Baani Singh, Kirti Sharma, Nitin Kumar, Deepinder Singh, Sarabjit Mastana

**Affiliations:** 1Department of Human Genetics, Punjabi University, Patiala 147002, India; singhmonica2017@gmail.com (M.S.); baani_rs22@pbi.ac.in (B.S.); kirti_rs22@pbi.ac.in (K.S.); nitin_rs19@pbi.ac.in (N.K.); 2VardhmanMahavir Health Care, Urban Estate Ph-II, Patiala 147002, India; deepinder.deepak@gmail.com; 3Human Genomics Laboratory, School of Sport, Exercise and Health Sciences, Loughborough University, Loughborough LE11 3TU, UK; s.s.mastana@lboro.ac.uk

**Keywords:** bone remodeling, signaling pathways, receptor activator of nuclear factor kappa-B, (RANK)-osteoprotegrin (OPG) and RANK ligand, macrophage colony-stimulating factor, Wnt/β-catenin, Jagged/Notch, bone morphogenetic protein

## Abstract

Bone remodeling is an intricate process executed throughout one’s whole life via the cross-talk of several cellular events, progenitor cells and signaling pathways. It is an imperative mechanism for regaining bone loss, recovering damaged tissue and repairing fractures. To achieve this, molecular signaling pathways play a central role in regulating pathological and causal mechanisms in different diseases. Similarly, microRNAs (miRNAs) have shown promising results in disease management by mediating mRNA targeted gene expression and post-transcriptional gene function. However, the role and relevance of these miRNAs in signaling processes, which regulate the delicate balance between bone formation and bone resorption, are unclear. This review aims to summarize current knowledge of bone remodeling from two perspectives: firstly, we outline the modus operandi of five major molecular signaling pathways, i.e.,the receptor activator of nuclear factor kappa-B (RANK)-osteoprotegrin (OPG) and RANK ligand (RANK-OPG-RANKL), macrophage colony-stimulating factor (M-CSF), Wnt/β-catenin, Jagged/Notch and bone morphogenetic protein (BMP) pathways in regards to bone cell formation and function; and secondly, the miRNAs that participate in these pathways are introduced. Probing the miRNA-mediated regulation of these pathways may help in preparing the foundation for developing targeted strategies in bone remodeling, repair and regeneration.

## 1. Introduction

The rising number of bone diseases along with skeletal problems and rheumatological conditions are severely impacting human health globally [1]. Affiliated with pain and agony, bone diseases may cause compromised movement/motion, bone deterioration and degradation, fracture risk and bone death, if timely curative management is delayed or denied. A highly mineralized tissue, bone has an inherent capacity to repair and regenerate itself, reversing bone loss. Bone remodeling is a process in which several cellular events are directed to replace damaged and dying bone with new bone. Many physical and physiological therapies are in place along with medications, diet control and lifestyle modifications, but the prevalence of several bone diseases (if not all) is increasing, which has prompted researchers and clinicians to look for newer and better methods to curtail bone diseases [2].

The global statistics on bone health are not at all encouraging and are rather alarming. Deteriorating bone conditions due to impaired remodeling and low bone mass are contributing substantially to years of life lost due to disability (YLDs) and years of life lost (YLLs) on a global scale [3]. Data from 204 countries have shown 178 million new cases of fractures, 455 million acute or chronic fractures along with 25.8 million YLDs from 1990 to 2019, which severely impacts the musculoskeletal well-being of people [4]. The estimation of the global burden of disease in more than 150 musculoskeletal conditions, including bones, joints, cartilage, muscles, the spine, ligaments and tendons, has highlighted that 494 million individuals are suffering from musculoskeletal disorders. These numbers are projected to increase at startling rates in the upcoming years [1].In the midst of this global situation surrounding bone health, prioritizing effective strategies for bone repair is of immense importance. Moreover, a considerable number of these impinging scenarios are non-modifiable or at least difficult to reverse; hence, scientists are investigating and sifting through several of the molecular pathways involved to identify novel elements that can be used as therapeutics for bone remodeling, repair and regeneration.

Commonly used strategies, including pharmacotherapy, physical therapy, orthopedic management, lifestyle modifications, nutritional interventions, clinical therapies, pain management and surgical procedures, have partially resolved associated inflammation, damaged tissue and bone loss [5,6,7]. Therapeutic approaches, including mesenchymal stem cell replacement and pulsed electromagnetic field therapy, have also contributed to reducing the effect of clinical variables, which exacerbate bone pathologies [8,9]. In the face of evolving challenges of bone disorders, improved methodologies to curb and cure them are direly needed. Double-stranded RNA-mediated interference (RNAi)-based gene regulation controlling strategies for microRNA (miRNA) and small interfering RNA (siRNA) have shown promising results for bone remodeling [10,11,12]. MicroRNAs are epigenetic mediators of the post-transcriptional repression of gene function and participate in an array of cellular events, including cell proliferation, cell differentiation, cell apoptosis, metabolism, embryogenesis, organogenesis and angiogenesis. They also play significant roles in regulating skeletal homeostasis and modulate immune response to various types of stimuli. Additionally, scientists are trying to focus on the molecular pathways responsible for either osteogenesis or clastogenesis, so that effective strategies are devised to support and promote the balance between bone formation and bone resorption. The present review aims to assimilate all the relevant knowledge of miRNA’s role in bone remodeling and five major bone signaling pathways influenced by miRNAs to develop plans for harnessing them in bone remodeling.

## 2. MicroRNA Biogenesis

Since its discovery in *Caenorhabditis elegans* by Lee et al. in 1993, miRNA has been acknowledged both as a potential target as well as a multifaceted tool for gene therapy [13]. miRNAs are small non-coding ribonucleic acids (RNAs) that are 22 nt in length and function through an evolutionarily conserved phenomenon that regulates gene expression called post-transcriptional gene silencing (PTGS), also known as RNA interference (RNAi) [14]. The canonical pathway of miRNA biogenesis is initiated in the nucleus with the transcription of miRNA genes by the enzyme RNA polymerase II (Pol II) followed by 5′ capping and 3′ polyadenylation into double-stranded RNA (dsRNA) molecules exhibiting a short-hair pin loop structure known as pri-miRNAs, which are several kilobases long [15]. These pri-miRNAs undergo processing by a multi-protein complex known as a microprocessor, which is present in the nucleus. This microprocessor assembly is composed of two main proteins: the dsRNA-specific endoribonuclease (RNase) type III Drosha and the dsRNA-binding protein (dsRBP) Di George critical region 8 (DGCR8; human homolog of Pasha), which are the catalytic and non-catalytic subunits of the complex, respectively [16]. The DGCR8 subunit of the microprocessor complex binds to the apical loop (>10 nt) of pri-miRNAs and stabilizes the interaction between the Drosha and the 11 bp basal stems of pre-miRNAs, which leads to the cleavage of flanking regions of the pre-miRNAs by Drosha to form 60–90 nt long single hairpin dsRNAs known as premature or precursor miRNAs (pre-miRNAs) in the nucleus [15].

For their further processing, these pre-miRNAs are exported to the cytoplasm. Once the pre-miRNAs are released into the cytoplasm, the loop region of pre-miRNAs is cleaved by a macromolecular complex [17]. This macromolecular complex comprises two main proteins, an evolutionarily conserved helicase with type III endo-RNase activity known as Dicer and a dsRBP trans-activation response element (TAR) RNA-binding protein (TRBP; TARBP2), which are the catalytic and accessory subunits of the complex, respectively [18]. The TBRP subunit binds the pre-miRNAs and assists the interaction of the dsRBD domain of Dicer with pre-miRNAs through its dsRBD C domain followed by the binding of the PPC (Platform-PAZ-connector; PAZ: Piwi-Argonaute-Zwille) domain of Dicer with 2 nt 3′ overhang and 5′ overhang of the pre-miRNAs through its 5′ and 3′ pockets. This leads to the cleavage of pre-miRNA loops by the two RNase domains (RNase I and II) using 3′ and 5′ counting rules to form 22 nt long dsRNA duplexes consisting of the stem regions of the pre-miRNAs [19,20,21].

The non-canonical pathway of miRNA synthesis involves the formation of mature miRNA molecules by surpassing the major steps of the canonical pathway, i.e.,pri-miRNA cleavage by Drosha and pre-miRNA cleavage by Dicer, dividing them into two major pathways, i.e., Drosha-independent and Dicer-independent pathways, respectively. In the Drosha-independent pathways, the short introns with hairpin-forming tendencies form pre-miRNAs via splicing and intron-lariat debranching, whereas the processing of snoRNAs, shRNAs and pre-tRNA transcripts by the direct action of Dicer forms pre-miRNAs, which are further processed in a similar way as the canonical pathway to form maturemiRNAs [22]. One of the prominent Dicer-independent pathways exhibits identical steps for miRNA genesis as those of the canonical pathway till the export of the pre-miRNAs in the cytoplasm, and the pre-miRNAs are then cleaved directly by Argonaute (Ago2) protein to form mature miRNAs. Another Dicer-independent pathway involves the action of the tRNase Z enzyme on the precursor tRNA to release its 3′ trailer sequence fragment, which can act as mature miRNAs [22].

The final step of miRNA maturation is the incorporation of the dsRNA duplex (consisting of a guide strand and a passenger strand) into the RNA-induced silencing complex (RISC), resulting in the activation of the RISC through the conversion of dsRNA to single-stranded RNA (ssRNA) by Ago proteins. The RNA-induced transcriptional silencing complex (RITSC/RISC) is a multi-ribonucleoprotein complex formed by four major domains: the exonuclease domain (EXO), endonuclease domain (ENDO), domain containing homology-searching activity related to *E. coli* (RecA) and helicase domain. The EXO, ENDO and RecA are dsRNA binding domains with nucleolytic activity, and the helicase domain is formed by Ago proteins, which are responsible for the ATP-dependent RNA helicase activity of the RISC complex. The consequence of RISC loading is observed as the unwinding of the dsRNA duplex (22 bp) into a single strand (guide/seed strand; 22 nt) accompanied by the degradation of passenger strands, ultimately establishing the activated microRNA-RISC (miRISC) complex, which is capable of miRNA-mediated gene silencing [15].

## 3. MicroRNAs Inhibiting Osteogenesis

The level of sequence complementarity among miRNAs and target mRNAs governs the way gene silencing is performed by miRNAs. In the case of complete complementarity, miRNAs augment target mRNA decay, but they cause translational repression when there is partial complementarity [23]. The core of the miRISC complex is formed by the AGO protein, which is bound by miRNAs to the mRNA targets. The AGO protein binds to GW182 (TNRC6) protein (a 182 kDA glycine/tryptophan repeat-containing protein), which interacts directly with the carbon catabolite repression-negative on TATA-less (CCR4-NOT) transcription complex subunits 1 and 9 (CNOT1 and CNOT9) to recruit the CCR4–NOT complex [24]. The poly-A-specific ribonuclease subunit 2/3 (PAN2/PAN3) deadenylase complex is recruited by GW182’s interaction with poly-A-specific ribonuclease subunit 3 (PAN3) [24]. This, in turn, causes mRNA deadenylation via the coordinated actions of the PAN2/PAN3 complex and the deadenylase module of CCR4–NOT, which is made up of the CCR4-NOT transcription complex subunit 7/8 (CNOT7/8) and CCR4-NOT transcription complex subunit 6/6L (CNOT6/6L subunits). The CNOT1-middle portion of the eIF4G (MIF4G) domain is essential for the de-capping factors and deadenylase module recruitment. By interacting with components involved in the translational repression of the target mRNA, the MIF4G domain further encourages translational repression [25]. The eukaryotic translation initiation factor 4A-2 (eIF4A2) displaces CNOT1 from the Eukaryotic initiation factor 4F (eIF4F) complex and prevents the ribosomal scanning of 5eIF4 through its interaction with the MIF4G domain. Through competition for binding to the MIF4G domain, eIF4A pushes DEAD box RNA helicases p54 (DDX6) out of CNOT1, preventing the recruitment of the deadenylase module to CNOT1 and halting mRNA decay. After DDX6 binds to the MIF4G domain in CNOT1, it interacts with the eIF4F binding protein (4E-T), which then inhibits translation or recruits eIF4F homologous protein (4EHP) and cap-binding protein, inhibiting the eIF4F complex by pushing eIF4E out of the cap. The recruitment of GRB10-interacting GYP protein 2 (GIGYF2) to miRISC is made possible by its interaction with the PPGL motif on GW182 through the GYF motif. GIGYF2 interacts directly and differently with CNOT1, CNOT7 and 4EHP to control translational repression [26,27]. miRNA-induced 5e→3′ mRNA decay requires the recruitment of the mRNA decapping enzyme 2 (DCP2) and its related protein, mRNA decapping enzyme 1 (DCP1), by CNOT1. The DDX6 protein functions as a hub for interactions with multiple other proteins, including PAT1 homolog 1, processing body MRNA decay factor (PATL1) and enhancer of mRNA-decapping protein3/4 (EDC3/4), which improve DCP1/2’s decapping activity. After the cap is removed, the mRNA is degraded by 5P→3′ exo-ribonuclease1 (XRN1). The co-interaction of the translation repression and mRNA degradation ultimately leads to the gene silencing of a variety of mRNA targets [26,28].

## 4. Regulation of Bone Remodeling by miRNAs

Bone remodeling is an intricate process that involves several signaling pathways and their interaction with growth promoters, bone cells, hormones, minerals and proinflammatory and anti-inflammatory mediators [29]. Bone remodeling ensures that mature bone is removed by osteoclasts (bone-resorbing cells), and new bone tissue is generated by osteoblasts (bone-forming cells), hence maintaining the balance between osteoclastogenesis (bone resorption) and osteogenesis (bone formation). The process of remodeling is under strong genetic control and any alteration in gene function can dysregulate it [30]. Even the slightest disturbance in the process of remodeling can trigger uncontrolled events of bone tissue degradation, the formation of osteoid (the unmineralized part of a bone matrix) and fibrosis (the thickening of tissue) [31].

miRNAs are important epigenetic managers which regulate gene expression by targeting mRNA and suppress post-transcriptional gene function. More than 15% of human genes encode miRNAs, and the majority of these miRNAs contribute to mRNA targeted translation repression [32]. Prevailing evidence indicates that miRNA-mediated gene regulation is crucial for bone homeostasis. These miRNAs are implicated in bone repair and regeneration as key players of bone remodeling through the regulation of osteogenesis, osteoclastogenesis and bone homeostasis [33,34,35].The latest clinical research has suggested that novel methods for accelerating bone formation are essential for recovering bone loss, managing bone disorders and resolving fractures [36,37].

The role and relevance of miRNAs in bone health have been unraveled from three cellular directions. Firstly, the miRNAs that were found to be involved in disrupting bone remodeling by blocking mRNA translation have been investigated [38]. Secondly, interactions of novel drugs with miRNAs have been examined in experimental knockdown animal models [39]. Thirdly, the development of anti-miRs, which inhibit the binding of miRNA with its target mRNA and hinders the formation of RNA-induced silencing complex (RISC),prevents the silencing of gene expression in a particular signaling pathway [40,41]. A comprehensive analysis of several miRNAs contributing to bone remodeling by targeting certain culprit genes has not yet been performed.

To comprehend intricate interactions and complex pathways involved in bone formation and bone degradation, understanding signaling pathways is a clinical prerequisite for the identification of molecular perpetrators inducing bone resorption and degradation as well as therapeutic targets for bone remodeling. Some of the major signaling pathways involved are receptor activator of nuclear factor kappa-B (RANK)-osteoprotegrin and RANK ligand (RANK-OPG-RANKL), macrophage colony-stimulating factor (M-CSF), Wnt/β-catenin, Jagged/Notch signaling and bone morphogenetic protein (BMP) pathways, which are genetically regulated by miRNAs [38].

Mesenchymal stem cells (MSCs) and their differentiation into bone cells contribute crucially to either bone resorption or bone remodeling. Several signaling pathways are involved in differentiating and maturing MSCs into cell types, such as osteoblasts (bone cells), chondrocytes (cartilage cells), myocytes (muscle cells) and adipocytes (adipose cells). Osteoblast precursors are derived from mesenchymal stem cells, which mature to osteoblasts in a Wnt-dependent canonical or a Wnt-independent non-canonical manner. RANKL and M-CSF produced by osteoblasts play a significant role in proliferation and bone resorption, whereas OPG inhibits osteoclastogenesis.

A deeper understanding of the cross-talk among bone cells, miRNAs and signaling pathways may reveal novel strategies to regulate bone remodeling. For instance, any miRNA that is important for targeting the bone resorption pathway can be transported to the fracture site through exosomal cargos for bone remodeling, as has been seen in transporting extracellular vesicles (exosomes) containing miRNA-21 from patients with traumatic brain injuries to repair the fracture site in a murine model [42]. The latest version of RNA Editing of Pri-miRNA for Efficient Suppression of miRNA (REPRESS) can edit pri-miRNA to reprogram stem cells to form a balance between bone resorption and bone formation. miRNAs participating in different stages of bone remodeling are shown in Table 1.

### 4.1. RANK-OPG-RANKL Pathway

Since the discovery of the RANK-OPG-RANKL pathway as a significant regulator of bone remodeling [74,75], its contributions to both bone formation (osteoblastogenesis) and bone resorption (osteoclastogenesis) have been revealed [76,77,78], making it a potential drug target. The RANK-OPG-RANKL pathway involves three interacting members whereby RANK (also known as TRANCER, ODFR and TNFRSF11A) and RANKL (TRANCE, OPGL, ODF and TNFSF11) are members of the tumor necrosis factor (TNF) superfamily. RANK, which is expressed by osteoclasts and dendritic cells, is a signaling receptor for RANKL. RANKL binds and activates RANK, which is the chief controller of osteoclastogenesis. Together, they support the activation of the cellular promoter nuclear factor of activated T cells 1 (NFATc1), which is a key precursor for the differentiation of mononuclear cells into osteoclast progenitors, as well as the upregulation of the production of cathepsin K, integrin β-3 (IGB3), calcitonin receptor (CT) and tartrate-resistant acid phosphatase (TRAP) [79]. OPG (also known as TNFRSF11) is a soluble protein secreted by stromal cells, which belongs to the TNF receptor superfamily. It operates as a decoy receptor which communicates with RANK and RANKL to block their binding, resulting in the inhibition of osteoclastogenesis [80]. Therefore, RANK and RANKL are indispensable for the regulation of osteoclast formation and their survival, whereas OPG is vitally important for protection against bone resorption and bone mass loss (Figure 1). Thus, relative levels of RANKL, OPG and the RANKL/OPG ratio are good indicators of bone mass and vitality, demonstrating a balance between bone formation and bone resorption.

Several reports have confirmed the significant role of various miRNAs in the arena of RANK-OPG-RANKL-pathway-mediated bone remodeling (Table 2). It has been observed that miR27a plays a role in osteoclastogenesis rather than osteoblastogenesis and hence is involved in bone-resorption-mediated osteoporosis [81]. miR27a participates in osteoclast differentiation through the mediation of sequestosome 1 (Sqstm1/p62), which has been implicated in RANK-mediated osteoclastogenesis in Paget’s disease. Davis et al. [82] has demonstrated that miR21 communicates with RANKL and high-mobility group box-1 (HMGB1) to decrease connexin 43 (Cx43), a cell signaling molecule responsible for cell differentiation and bone growth. The results of the study exhibited that the Cx43/miR21/HMGB1/RANKL pathway contributes to the protection of osteoclast apoptosis, hence promoting osteoclastogenesis. Wong et al. [83] demonstrated that miR21 manifests during osteoclastogenesis in RANKL-induced bone resorption. Sugatani et al. [84] highlighted that RANKL-induced osteoclastogenesis is regulated by miR-21, which targets programmed cell death 4 (PDCD4) protein levels. Another study demonstrated that estrogen downregulates miR-21, thereby increasing FasL and Caspase 3 protein levels in RANKL-induced osteoclastogenesis. The results of the study suggested that estrogen directly targets osteoclasts, and the ERα pathway is mediated by miR-21 levels [85]. The results of the study concluded that miR21 targets phosphatase and the tensin homolog (PTEN) gene and promotes osteoclastogenesis. Mizoguchi et al. [86] reported that the miR-31 target RhoA regulates cytoskeletal organization and bone resorption. Another study highlighted that estrogen receptor alpha (ERα) and miR-503 are highly expressed in osteoclastogenesis [45]. The data of the study suggested that oleanolic acid and antagomiR-503-5p block miR-503 expression, which is a novel strategy for inhibiting RANKL-induced osteoclast differentiation. miR155 has been shown to be upregulated in osteoporosis, in which it targets the leptin receptor (LEPR) [87]. Downregulating miR-155 activates the AMP-activated protein kinase (AMPK), which, in turn, attenuates the RANK-OPG-RANKL signaling axis, promoting bone formation. Another study reported that miR-106b is implicated in inflammatory-pathway-associated periprostheticosteolysis (PPO) in an experimental rat model [88]. The findings of the study demonstrated that the inhibition of miR-106b by the lentivirus-mediated miR-106b inhibitor significantly reduced inflammation and blocked the RANKL and PTEN/P13K/AKT signaling pathway.

Paget’s disease of bone (PDB) is characterized by dysregulated bone remodeling with excessive bone resorption. A study investigating miRNA profiles has suggested that miR-133a-3p is a positive regulator, whereas miR-155-5p and miR-146a-3p are negative regulators of osteoclastogenesis by affecting the p38MAP-kinase and RANKL pathways [48]. Liu et al. [49] examined the effect of isobavachalcone (IBC), a formulation derived from the Chinese herb Psoraleacorylifolia Linn, on the inhibition of osteoclastogenesis for the treatment of osteoporosis [49]. Another study has revealed that miR-155 target purine-rich binding protein 1 (PU1) and microthalmia-associated transcription factor (MITF) downregulate RANKL-induced osteoclast differentiation and hence play an inhibitory role in osteoclastogenesis [50]. It was observed that IBC suppressed RANKL-induced osteoclastogenesis in bone-marrow-derived monocytes/macrophages without any cytotoxicity. Chen et al. [89] reported that miR-503 is significantly reduced in circulating progenitors of osteoclast-Cd14+ peripheral blood mononuclear cells (PBMCs) in postmenopausal osteoporosis. The silencing of miR-503 in CD14+ PBMCs triggers osteoclastogenesis through the RANKL pathway, promotes bone resorption and decreases bone mass in ovariectomy mice. Another study reported that miR-214-3p mediates osteoclast activity and bone resorption in osteolytic bone metastasis (OBM) in breast cancer through the RANKL pathway [53]. It was reported that miR-214 inhibits osteoblastogenesis by targeting the activating transcription factor 4 (ATF4) gene and exhibits osteoclastogenesis through macrophage colony-stimulating factor (M-CSF) and the RANKL pathway [90]. The results suggested that miR-214 plays an important role in the differentiation of osteoclasts and can be a potential therapeutic target for osteoporosis. To understand the role of miRNA-34a in RANKL-induced bone resorption and osteoclastogenesis in osteoporosis, it was revealed that targeting miR-34a by transforming growth factor beta-induced factor 2 (Tgif2) increased bone resorption, suggesting it to be a key osteoclast suppressor and skeletal processor [52]. Another study exposed a novel biomarker, miR-182,that promotes inflammatory osteoclastogenesis driven by the RANKL pathway, indicating a better therapeutic strategy to prevent inflammatory osteoclastogenesis and bone resorption by targeting miR-182 [91]. Evidence suggests that miR-182 target heme oxygenase-1 (HO-1) upregulates RANKL-induced osteoclastogenesis in inflammatory episodes in rheumatoid arthritis (RA) patients. The inhibition of miR-182 was also observed to downregulate RANKL-induced bone resorption and osteoclastogenesis in RA monocytes by targeting double-stranded RNA-dependent protein kinase (PKR) and mediating beta interferons(IFN-β) [64,92].The administration of miR-135b into exosomes derived from mesenchymal stem cells targeted PDCD4 mRNA to mitigate bone loss, hence ameliorating osteonecrosis of femoral head (ONFH) severity in a rat model [46]. A study exhibited the role of the miR-17/20a cluster in the suppression of glucocorticoid-mediated osteoclast differentiation through the silencing of dexamethasone-stimulated RANKL gene expression in osteoblast cells [93]. Another study revealed that miR-106b is a novel suppressor of osteolysis and osteoclastogenesis by inhibiting the RANKL pathway in giant cell tumor stromal cells [94]. Ohnuma et al. [51] demonstrated the retardation of TNF-α- and IL-6-mediated osteoclast differentiation by targeting NFATC1 expression. Another study reported the recombinant adeno-associated viral vector-mediated delivery of miRNA 34a-5p, which resulted in improved bone mass with limited side effects on non-skeletal tissues. This miRNA targets TGIF2 in osteoclasts and NOTCH1 in osteoblasts to ameliorate both post-menopausal and senile osteoporosis [52]. A study by Ma et al. [95] provided evidence that the therapeutic effect of Icariin treatment for glucocorticoid-induced osteoporosis is partially mediated by the activation of the miR-186-directed repression of CTSK expression, which alleviated the Dex-stimulated disequilibrium of calcium homeostasis.

### 4.2. M-CSF Signaling Pathway

Macrophage colony-stimulating factor (M-CSF) is a hematopoietic growth factor, which stimulates the proliferation, differentiation and activation of monocyte lineages and regulates their survival. M-CSF is a ligand present on membrane for colony-stimulating factor 1 receptor (CSF1R or c-Fms), which organizes the differentiation of myeloid lineage cells, including macrophages/monocytes, osteoclasts, microglia, Langerhans cells and Paneth cells. M-CSF signaling interacts with RANKL to induce osteoclast proliferation and differentiation from precursor cells [96]. M-CSF/CSF1R signaling communicates with RANK, MAPK and NFATC1 signaling to induce c-Myc and c-Fos expression, leading to the induction of metabolic reprogramming and osteoclastogenesis [97,98]. M-CSF signaling also influences fracture healing and repair by recruiting stem cells to the fracture site for the development of hard calluses by inhibiting osteoclastogenesis [99].

Very few reports are available on miRNAs participating in bone remodeling through M-CSF signaling (Table 3). It was reported that miR-7b inhibits osteoclastogenesis by regulating NFATC1, c-Fos, MAPK, Akt and TRAF6, suggesting that miR-7b can be a therapeutic marker for the treatment of osteoclastogenesis-related bone diseases [100]. Another report revealed that miR-214 is expressed in osteoclastogenesis through M-CSF and RANKL signaling, suggesting it to be a therapeutic target for osteoporosis [93]. miR-155 was reported to regulate osteoclastogenesis through the stimulation of M-CSF and RANK [101]. Mao et al. discovered that inhibiting miR-155 upregulates LEPR via several pathways, including M-CSF, resulting in suppressed osteoclastogenesis and bone resorption in osteoporotic mice. Another report demonstrated that miR-21 attenuates the production of osteoclasts induced by M-CSF signaling in juvenile idiopathic arthritis [102]. It was confirmed that miR-143-3p inhibits osteoclast differentiation in synovial fibroblasts by the downregulation of the M-CSF, RANKL and MAPK pathways in arthritic rats [103]. A study has suggested the role of miR-125a in stimulating osteoclast differentiation by regulating both RANKL and M-CSF signaling through its target TRAF6, which can be employed to manage metabolic disease [104]. Another study also showed that miR-146a-3p is downregulated in PDB patients, which is manifested as reduced bone resorption by osteoclasts in response to the upregulation of its target, TRAF6 [48]. It is observed that miR-195a suppressed both RANKL and CSF1 gene expression, which resulted in halted osteoclast formation and activity in OVX-induced osteoporosis [105]. Another study exhibited the stimulatory role of miR-346-3p in osteoclastogenesis by targeting the TRAF3 gene, which can be harnessed in developing treatments for bone-loss-associated pathologies [106]. One of the molecular mechanisms for regulating osteoclast formation and differentiation is influencing the levels of M-CSFR by miR-223 [107].

### 4.3. Wnt/β-Catenin Signaling Pathway

The Wnt gene was observed for the first time as a retroviral insertional mutation in mouse mammary tumor virus at the integrase-1 (intl-1) gene, which later on was identified as a mouse homolog of the wingless gene (wg) [108,109]. Because the functional properties and translated proteins of both of these genes are similar, scientists combine parts of their name to form the Wnt gene. Wnt signaling is expressed in canonical and non-canonical pathways. Canonical Wnt signaling is called the Wnt/β-catenin pathway, as it involves the nuclear translocation of β-catenin along with the activation of target genes through transcription factors including T cell factor/lymphoid enhancer factor (TCF/LEF). Non-canonical Wnt signaling does not involve β-catenin, TCF/LEF or Wnt planar cell polarity [110]. The functions of the canonical Wnt pathway include controlling cell proliferation, whereas the non-canonical mode of Wnt signaling regulates cell polarity and migration. Both of these canonical and non-canonical pathways interact to play a significant role in embryonic development, the self-regeneration of tissue, liver tissue development, lung tissue regeneration, osteoblast maturation and the maintenance of their activity [111,112,113]. Wnt/β-catenin signaling has four components, which are involved in its effective expression. Firstly, the extracellular signals which are sensed by Wnt proteins comprise Wnt1, Wnt3a and Wnt5a. Secondly, the cell surface/membrane component is regulated by Wnt receptors comprising the seven-fold transmembrane receptor Frizzled (FZD) and low-density lipoprotein receptor 5 and 6 (LRP5/6). Thirdly, the cytoplasmic component is mediated by β-catenin, glycogen synthase kinase-3β (GSK-3β), disheveled (DVL) protein, axis inhibition (AXIN), adenomatous polyposis coli (APC) and casein kinase 1 (CK1). Lastly, the nuclear component includes β-catenin, which is translocated to the nucleus along with matrix metalloproteinases (MMPs) and cellular MyC (c-MyC) [114]. Combining these four components, canonical Wnt signaling is activated by recognizing the extracellular Wnt ligand by the membrane receptor, which induces β-catenin and its translocation in the nucleus, finally triggering the expression of genes responsible for cell proliferation, survival, vitality, differentiation and migration [115]. However, in the absence of Wnt signaling, β-catenin is degraded along with its receptors, and in the activation of Wnt signaling, β-catenin binds to its receptor AXIN/LRP, which evades the degradation of β-catenin and is translocated to the nucleus for the activation of target genes [116].

The Wnt signaling pathway interacts with the RANK-OPG-RANKL pathway to suppress its bone resorption potential. The non-canonical Wnt pathway enhances RANKL-induced osteoclastogenesis. Therefore, Wnt signaling promotes both bone formation and bone resorption [117]. Mutations in Wnt signaling may lead to several pathological conditions including impaired bone repair, malignant osteoid generation and autoimmune conditions, suggesting its role in skeletal tissue regeneration, bone growth, bone repair and healing [118]. Genetic and pharmacological manipulations of the Wnt receptor LRP-5 and its antagonist, sclerostin has, exhibited that Wnt signaling plays a central role in osteogenesis and bone remodeling [119].

The potential role of known microRNAs targeting the Wnt/β-catenin pathway has been shown in Table 4. Osteonectin is essential for the recovery of bone loss and bone remodeling. miR-29a and miR-29c were reported to interact with osteonectin for its downregulation of osteonectin protein levels, preventing matrix maturation and mineralization by activating canonical Wnt signaling [71]. The transient modulation of osteo miR-199b, miR-1274a and miR-30b by the targeting of bone morphogenetic protein receptor (BMPR), T cell factor (TCF) and the suppressor of mothers against decapentaplegic (SMAD) protein increases Wnt-signaling-induced osteogenesis [54]. Another study highlighted that the effects of glucocorticoid (GC)-induced impaired Wnt signaling can be limited by miR-29a and improve osteoblastic differentiation and mineralization [120]. The results of the study suggested that miR-29a can be used for alleviating GC-induced bone deterioration. miRNA expression profiling in the reparative phase of the osteonecrosis of femoral head (ONFH) was investigated, which revealed that miR-34a and miR146a were upregulated in the Wnt-signaling-induced reparative phase in ONFH [121]. Another study demonstrated that miR-29a mediates tumor necrosis factor-α (TNF-α) induced bone loss by targeting Dickkopf Wnt signaling pathway inhibitor 1 (DKK1) and GSK-3β, hence activating Wnt signaling for the inhibition of osteogenesis [122]. It was reported that miR-141 and miR-22 are negative regulators of Wnt/β-catenin signaling and osteogenesis [66]. miRNA and mRNA expression profiles from the synovial tissue of mice with rheumatoid arthritis were examined, which highlighted that the secretion of miR-221-3p in exosomes was upregulated in synovial fibroblasts treated with proinflammatory TNF-α. This overexpression of miR-221-3p decreases calvarial osteoblast differentiation and tissue mineralization by targeting Wnt and BMP signaling [64]. It was observed that miR-29a expression is positively correlated with canonical Wnt signaling in ankolysing spondylitis [123]. Another study reported that the overexpression of miR-335-5p in osteoblast lineage cells promotes osteogenic differentiation and bone growth in mice by downregulating the Wnt antagonist DKK1 [124].

Excessive levels of endogenous GC upregulate the expression of the Wnt antagonists sclerostin and DKK1 and are associated with hypercortisolism in humans. This downregulation of Wnt signaling suppresses osteoblast differentiation and osteoclastogenesis [68]. This process dysregulates the expression of miR-210-5p, miR-211, miR-135a-5p and miR-204-5p in patients with Cushing’s disease. Similarly, it was reported that excessive GC increases the expression of the Wnt signaling antagonists DKK1 and Wnt10B and dysregulates the expression of miR-199a-5p, which is involved in MSCs’ commitment to chondrocytes in active acromegaly [125]. miR-185 was observed to inhibit osteoblastogenesis in fracture healing by targeting the parathyroid hormone (PTH) gene and downregulate Wnt/β-catenin signaling, suggesting that the suppression of miR-185 is an effective strategy to promote osteoblast growth and proliferation in fracture healing by activating Wnt/β-catenin signaling [43]. Sui et al. [126] used a biodegradable lipidoidnano-carrier for the delivery of miR-335-5p to cells to enhance osteogenesis in vitro and calvarial bone healing in vivo. The transfection of C3H10T1/2 cells and BMSCs with lipidoid-miR-335-5p downregulated the expression of DKK1, thereby increasing osteogenesis. The results of the study suggested that the lipidoid delivery of miRNAs to induce osteogenesis and bone regeneration can be a therapeutic strategy. In order to understand the role of miRNAs in the bioengineered reconstruction of bone grafts, miR-26a-5p and antimiR-26a-5p in adipose-derived mesenchymal stem cells (ADSCs) were combined with calcium phosphate(CP) scaffolds in a rat femur model [127]. The results revealed that miR-26a-5p attenuated the bone formation process, whereas antimiR-26a-5p accelerated bone formation via Wnt/(Ca_2_^+^) signaling, suggesting that antimiR-26a-5p-modified ADSCs withCB are better for constructing bone grafts for bone repair and regeneration. Another study exhibited that miR-200c regulates Wnt signaling for the promotion of osteogenic differentiation in hBMSCsand hence may serve as an efficient osteo-inductive agent for better bone repair [55]. Bone loss is aconsequence of ageing, leading to bone disorders such as osteoporosis and osteoarthritis. It was observed that miR-146a is an epigenetic regulator of age-related bone loss by mediating anabolic Wnt signaling [63]. The results of astudy suggested that targeting miR-146a can be an efficient strategy to prevent age-related bone loss and bone impairment. Another study suggested that piezoelectric microvibration stimulation (PMVS) intervention in osteoporosis may promote Peizo1, miR-29a and Wnt signaling to upregulate osteogenesis, downregulate osteoclastogenesis and recover estrogen-deficiency-related bone loss [128]. The results of another study indicated that miR-26b regulates subchondral bone loss by influencing osteogenic differentiation in BMSCs and activating Wnt/β-catenin signaling in temporomandibular joint osteoarthritis [56]. It has been explained that miR-433-3p promotes osteoblast differentiation by directly targeting WNT antagonist DKK1 in an OVX rat model [44]. Another study revealed that the activation of miR-335-5p mitigated diabetic osteoporosis by manifesting an anti-apoptotic effect on osteoblasts by decreasing DKK1 expression [129]. miR-218 is shown to coordinate a positive feedback loop of Wnt signaling by silencing the major three WNT inhibitors, i.e., SOST, DKK2 and SFRP2, which exacerbate both osteoblast differentiation and the osteomimicry of metastatic cells [130]. A study exhibited the role of miR-21 in determining the fate of MSCs by adversely modulating SOX2 expression by inducing osteogenesis and retarding chondrogenesis in spindle-shaped fetal MSCs and exerting the opposite effect in round-shaped fetal MSCs and adult bonemarrow MSCs [131]. Chen et al. [59] suggested that miR-125b aggravates senile osteoporosis by influencing osterix expression, which retrogrades the proliferative ability of hBMSCs and their commitment towards osteogenesis. Another study has also demonstrated that the cell fate of hBMSCs was determined by the regulation of osterix expression by miR-637 to maintain the balance between adipocytes and osteocytes, as it is observed to inhibit Wnt-mediated osteogenic differentiation [60].

### 4.4. Notch Signaling Pathway

Bone tissue is dynamic: it begins developing during embryogenesis and continues over one’s whole life, even after attaining maturation in adulthood [132]. The main players in this process of remodeling are osteoblasts and osteoclasts, which are guided and controlled by several signaling processes, and Notch signaling is one of these pathways. Notch signaling is a cell–cell communication pathway, which involves single-pass protein transmembrane ligands comprising Jagged 1 and2 and delta-like ligands (DLLs) 1, 3 and 4. These ligands facilitate the binding of Notch receptors (1–4) at the cell surface [133]. This ligand–receptor complex promotes the proteolytic cleavage of Notch receptors by the γ-secretase complex, releasing Notch intracellular domain (NICD) in the cytoplasm. After the translocation of this complex in the nucleus, it interacts with the DNA binding protein kappa J region RBP-jk/CBF1 and forms a complex comprising transcriptional co-activator mastermind-like 1–3 (MAML 1–3) and RBP-jkto activate Notch target genes, including transcriptional repressors such as hairy and enhancer of split (HES) protein along with YRPW protein (HEY). Under different physiological and pathological conditions, Notch signaling interacts with Wnt/β-catenin and BMP pathways for the regulation of bone remodeling [134]. Mutations in Notch or jagged genes have been implicated in several disorders in humans including Algille syndrome, Hagdu–Cheney syndrome, osteopenia and fractures [135,136]. It has been reported that canonical RBP-jk-dependent Notch expression controls the proliferation of mesenchymal progenitors and their differentiation during skeletal development [137], whereas the genetic ablation of Jagged 1 causes the downregulation of HES and HEY and causes severe abnormalities of trabecular bone volume and trabecular spacing [138]. Alluding to these findings, Notch signaling is an effective determinant of osteoblast differentiation, osteoclast recruitment, the proliferation of osteoblasts/osteoclasts, the mineralization of callus tissue, and osteoclast recruitment and regeneration after fractures [139,140].

M-CSF signaling along with RANK/RANKL contributes significantly to osteoclastogenesis, whereas OPG, TGFβ and Wnt/β-catenin inhibit it. MicroRNAs, i.e., miR-29a, miR-28-5p, miR-30, miR-133 and miR-199, contribute to these pathways for osteoclastogenesis. On the other hand, the differentiation of osteoblasts is triggered and supported by OPG, TGFβ and Wnt/β-catenin signaling, whereas RANK/RANKL and M-CSF inhibit this process. MicroRNAs, i.e., miR-21, miR-34a, miR-146a-3p, miR-155 and miR-27a, participate in the process of osteoblastogenesis (Figure 2). Mesenchymal stem cells (MSCs) have the ability to undergo multilineage differentiation into several connective tissue types, such as osteocytes, osteoblasts, adipocytes, chondrocytes and myoblasts. They can also undergo transdifferentiation to form neurons [141].

The potential role of knownmicroRNAs targeting the Notch/Jagged pathway is shown in Table 5.It was reported that miR-9 promotes the differentiation of bone MSCs into neurons by Notch signaling for the first time in Ref. [142]. Another report highlighted the role of miR-34 in the post-transcriptional regulation of Notch signaling for the differentiation of osteoclasts and demonstrated that miR-34 served as a critical regulator of osteoblastogenesis [70]. miRNAs have been identified as novel regulators of human stromal stem cells’ (hMSCs) differentiation. miR-34a was reported to be a regulator of hMSCs and osteoblastic differentiation by targeting Jagged 1, a ligand for Notch signaling [143]. The results of a study suggested that miR-34a blocks bone formation by inhibiting osteoblastic and hMSCs’ differentiation. The inhibition of Notch signaling in bone marrow cells upregulates the expression of miR-155, thereby increasing proinflammatory cytokine production [144]. Another study highlighted that miR-34a-5p is an important player in dexamethasone-inhibited BMSC proliferation and osteogenesis by the activation of Notch signaling [145]. Tian et al. [146] reported that miR-30a acts as a commendable player in MSCs’ differentiation into chondrocytes by inhibiting DLL4, a ligand for Notch signaling. The results of the study suggested the importance of miR-30a-regulated MSC therapy in cartilage-deteriorating disorders like osteoarthritis. Additionally, the miR-497~195 cluster has been found to mediate the coupling of angiogenesis–osteogenesis for the maintenance of bone loss through the activation of Notch signaling [147]. It has been observed that miR-146a expression is suppressed in osteoarthritic lesions of articular cartilage. Its role in post-traumatic osteoarthritis (PTOA) was investigated, which revealed that the chondrogenic overexpression of miR-146a by the Notch inhibitor decreases proinflammatory interlekin1 beta (IL-1β)-induced inflammation in joint degradation, suggesting it to be a therapeutic marker for inflammation-driven OA and PTOA [65]. Another study exposed that miR-487b-3p negatively controls osteoblastogenesis by blocking Notch-regulated ankyrin-repeat protein (Nrarp) expression, which further attenuates Runx-2 and Wnt signaling [148].

Mesenchymal stem cells are differentiated into osteoblasts which are transformed into osteocytes through the process of mineralization, which is facilitated by Notch activation (Figure 3). The differential mRNA expression profiles of bone marrow mesenchymal stem cells (BM-MSCs) of adolescent idiopathic scoliosis (AIS) patients were analyzed, which revealed that miR-17-5p, miR-15a-5p, miR-181b-5p, miR-106a-5p, miR-106b-5p, miR-16-5p and miR-93-5p play crucial roles in the osteogenic differentiation of MSCs through the interaction of multiple signaling pathways, including Notch [149].It was reported that the expression of homebox transcript antisense intergenic RNA (HOTAIR) plays an important role in the degeneration of nucleus pulposus (NP) cells in intervertebral disc degeneration (IVDD) [150]. The results of a study have demonstrated that HOTAIR sequesters miR-34a-5p and downregulates the apoptosis of NP cells by mediating Notch signaling. Another study reported that miR-342-5p downregulates the proliferation, migration and differentiation of osteoblasts by blocking Notch signaling [151]. It was revealed that miR-199b-5p mediates chondrogenic differentiation by targeting jagged 1 in cartilage-related disorders [58]. Recently, it was reported that miR-210 overexpression may improve the microstructure of bone tissue and reduce bone resorption in ovariectomized rats by activating VEGF/Notch1 signaling, suggesting its protective role in osteoporosis in post-menopausal rats [152]. Ina recent study, it was reported that the transplantation of miR-28-5p in BMSCs directly targets Notch1, which promotes functional recovery in patients with spinal cord injuries [153]. It was revealed that miR-199b-5p inhibits the ossification of ligamentum flavum cells by the downregulation of JAG1-associated Notch signalosomes [154]. An integrated miR-microarray and bioinformatics analysis showed that the dysregulation of various pathways controlled by different miRNAs are implicated in the pathogenesis of myelodysplastic syndromes involving the negative modulation of the Notch pathway by the overexpression of miR-195-5p by targeting the Notch ligand DLL1 [155]. Another study illustrated the effect of the prenatal exposure of dexamethasone in the form of osteopenia inherited by multiple generations in a rat model governed by the active silencing of JAG1/Notch1 signal transduction by miR-98-3p [69].

### 4.5. TGF-β/BMP Signaling Pathway

Bone morphogenetic proteins (BMPs) are a group of signaling molecules that are members of the transforming growth factor-β (TGF-β) superfamily of proteins [156]. These signaling proteins play a substantial role in skeletal development and post-natal bone homeostasis via the differentiation of osteoblasts and chondrocytes from mesenchymal progenitors and contribute to osteoclast fate [157]. TGF-β and BMPs interact with the tetrameric receptor complex and express signaling in canonical and non-canonical ways. In the canonical SMAD-dependent pathway, TGF-β/BMP ligands, their receptors and SMAD participate, whereas in the non-canonical SMAD-independent pathway, p38 mitogen-activated protein kinase (p38MAPK) participates in the pluripotent mesenchymal progenitor-derived differentiation of osteoblasts for skeletal health [157]. Both canonical and non-canonical pathways connect to same transcription factors, including Runt-related transcription factor 2 (Runx2) to induce osteogenesis. Both canonical and non-canonical pathways in TGF-β/BMP signaling are shown in Figure 4. TGF-β/BMP signaling is controlled by several factors including miRNAs, epigenetic modulators and the ubiquitin–proteasome complex. Knockdown/dysregulation in TGF-β/BMP signaling is implicated in several bone disorders in humans [158]. The TGF-β/BMP pathway interacts with other signaling trajectories, including Wnt/β-catenin, Notch, Hedgehog (Hh), RANK-OPG-RANKL and parathyroid hormone-related protein (PTHrP), to organize bone remodeling, repair and osteogenesis [159].

Several miRNAs have been indicated in TGF-β/BMP signaling-mediated bone remodeling (Table 6). It has been observed that miR-133 inhibits TGF-β/BMP-induced osteogenesis by targeting Runx2 and SMAD5 [62]. Another study reported the negative effect of miR-30 on osteoblastogenesis through the targeting of Runx2 and SMAD1 [160]. The synergistic signaling of transcription factors such as Runx2 and Osterix regulates osteogenesis. It has been reported that the overexpression of miR-322 increases BMP-2 via upregulating the expression of osterix and osteogenic genes [161]. Another study reported that miR-141 and miR-200a mediate BMP-2-induced pre-osteoblastic differentiation through the silencing of the distal-less homeobox 5 (DLX5) gene [162]. miR-542-3p was observed to suppress the proliferation and differentiation of osteoblasts by targeting BMP-7 [67]. Another study revealed that miR-20a enhances bone formation by triggering the osteogenesis of hMSCs by targeting peroxisome proliferator-activated receptor gamma (PPAR-γ), cysteine rich motor neuron 1 (CRIM1) and bone morphogenetic protein and activin membrane-bound inhibitor (BAMBI), the antagonists of BMP signaling [57]. Another chondrocyte-specific miR-140 targets the BMP inhibitor aspartyl aminopeptidase (DNPEP) to cause bone deformities in craniofacial and endochondral bone [163]. It was reported that the expression of miR-146a induces chondrocyte apoptosis by targeting SMAD4 [164]. BMP-2 responsive miR-199a adversely regulates chondrocyte differentiation by the direct targeting of SMAD1 [165]. The expression profiling of miRNA in chondrocyte differentiation revealed several miRNAs that express in osteoarthritic chondrocytes, including miR-20b, miR-345 and miR-146, which target TGF-β/BMP signaling [61,166,167]. Xie et al. investigated specific miRNAs in ADSCs during BMP-induced osteogenesis, which revealed that the overexpression of miR-146a suppressed ADSCs’ osteogenesis. The knockdown effect of SMAD4 downregulated the expression of miR-146a, resulting in ADSC osteogenesis, bone regeneration and repair in both in vitro as well as in vivo settings [168]. miR-433-3p was observed to be implicated in the negative regulation of bone formation targeting the Wnt/β-catenin, GC, MAPK and PTH signaling pathways [72]. The results of the study suggested that the local delivery of the miR-433-3p inhibitor can be an effective preventive mechanism against bone loss and bone defects. Mesenchymal stem-cell-derived extracellular vesicles (MSC-EVs) have the potential to treat osteoporosis. RNA sequencing exhibited that miR-21, miR-29, miR221 and Let-7a were overexpressed through BMP signaling in Wharton’s jelly MSC-EVs [169]. The results of the study suggested the therapeutic use of exogenous Wharton’s jelly-derived MSC-EVs in osteoporosis. Another study examined the role of miR-20a in fluid sheer-stress-associated osteogenic differentiation, which revealed that miR-20a increases fluid sheer-stress-induced osteogenesis by upregulating the BMP2 signaling pathway by targeting BAMBI and SMAD6 [170].The function of miR-145 as an inhibitor of osteoclastogenesis by restraining SMAD3 expression in an ovariectomized mice model was observed [73]. A study demonstrated the suppression of the synergistic effect of RUNX2 and SMAD5 by miR-133 and miR-135, respectively, which hampered the commitment of osteoprogenitors towards osteoblast lineage cells [62]. Wu et al. [171] indicated the role of miR-126-5p as a repressor of the formation of both multinucleated giant cell and osteolytic lesions by targeting MMP-13 expression in giant cell tumors. Another study has shown that the attenuation of RUNX2 expression by two microRNAs, i.e., miR-135 and miR-203, exhibited an anti-metastatic effect in breast cancer, which can be used as a gene therapy for bone metastasis disease [172].

## 5. Clinical Implications and Future Directions

It is beyond doubt that miRNA-based therapeutic strategies in several diseases have an astounding impact on genomic medicine, but their effective utility for the management of diverse bone disorders still needs to be addressed [35]. First of all, a crucial challenge is the complexity of miRNA-mediated regulatory networks with intricate cross-talk amongst miRNAs and their respective gene targets. Deciphering such processes comprehensively is essential for developing efficient therapeutic strategies [38]. Secondly, the translation of preclinical studies in animals for the clinical benefit of humans is required. For that, clinical trials in humans are essential for the rigorous validation of the safety and efficacy of miRNA-based targeted therapies. Furthermore, the interaction of several signaling pathways for inducing osteoblastogenesis or osteoclastogenesis creates a complex milieu of participating ligands, their receptors, cells and transcription factors [39]. Comprehending the context-dependent effects of the miRNAs regulating these pathways in different phases of bone remodeling is complex but obligatory for tailoring therapeutic interventions for a specific patient profile.

Signaling pathways, such as RANK-OPG-RANKL, Wnt/β-catenin, M-CSF, Notch and TGF-β/BMP, offer promising avenues for drug development. The knowledge obtained so far will not only help in the identification of newer drug targets and potential druggable targets, but also help in devising strategies for bone cell-level delivery targets in order to resolve bone remodeling [49,76,77,78,79,86,87,88]. For instance, monoclonal antibodies (denosumab) to inhibit RANKL for mitigating osteoclastogenesis were discovered due to a profound knowledge of the RANK-OPG-RANKL pathway [6]. Similarly, targets for the Wnt signaling pathway inhibitors Sclerostin and Dickkopf-1 may modulate and enhance osteoblast differentiation [114]. Similarly, targeting Notch signaling could improve MSC differentiation into bone-forming cells, and BMP agonists could stimulate bone regeneration [137,138,139,140]. Identifying specific miRNAs that inhibit dysregulated remodeling could lead to innovative treatments for several bone diseases. Overall, a deep understanding of these pathways and their cross-talk with bone cells and signaling molecules will be essential for developing personalized therapies in bone disorders [38,39].This review summarizes the current knowledge of miRNAs and their clinical implications related to signaling pathways from the perspective of bone remodeling, repair and regeneration. This harnessed knowledge may provide the foundation for developing future personalized therapies for correcting impaired bone remodeling. Nonetheless, further efforts should be made to devise effective strategies to deliver specific miRNAs at the site of damaged bone tissue. A deeper exploration of the cross-talk between miRNAs and signaling pathways, regulatory elements such as long non-coding RNAs and epigenetic mediators will contribute to a more comprehensive understanding of the regulatory landscape in bone remodeling. Overcoming challenges for decoding complex miRNA networks, translating preclinical knowledge to pragmatic clinical applications, identifying context-specific and tissue-specific miRNAs, recognizing targets in multiple pathways and developing efficient delivery methods of miRNAs are required and will be clarified by future studies for harnessing the full potential of miRNA-based mechanisms in correcting bone remodeling for bone repair and regeneration.

## 6. Limitations of the Study

The present study is an exhaustive account of the contribution of potential miRNAs to skeletal homeostasis and five major signaling pathways involved in bone remodeling; however, it has some limitations. This review involved the major pathways of remodeling; nonetheless, the participation of other signaling pathways, such as the hedgehog (HH), parathyroid hormone 1 receptor (PTH1R), fibroblast growth factor (FGF) and mitogen-activated protein kinase (MAPK) pathways, could have been included to make this review more informative. Furthermore, the relationship between miRNAs and signaling pathways, for which information is inadequate due to a dearth of validated sources, could be used to design drug targets.

## 7. Conclusions

This review has analyzed and offered the important aspects of miRNAs’ regulation of osteogenesis, cell differentiation and matrix mineralization through targeting the specific genes responsible for osteoclastogenesis and chondrogenesis. Current knowledge regarding miRNAs’influences on bone remodeling through five major signaling pathways, i.e., RANK-OPG-RANKL, M-CSF, Wnt/β-catenin, Notch and TGF/BMP, has been summarized. The clinical applications of miRNA therapeutics will be further clarified by upcoming studies on the aspects of clinical events and biological effectors involved in miRNA-influenced bone remodeling.

## Figures and Tables

**Figure 1 diseases-12-00252-f001:**
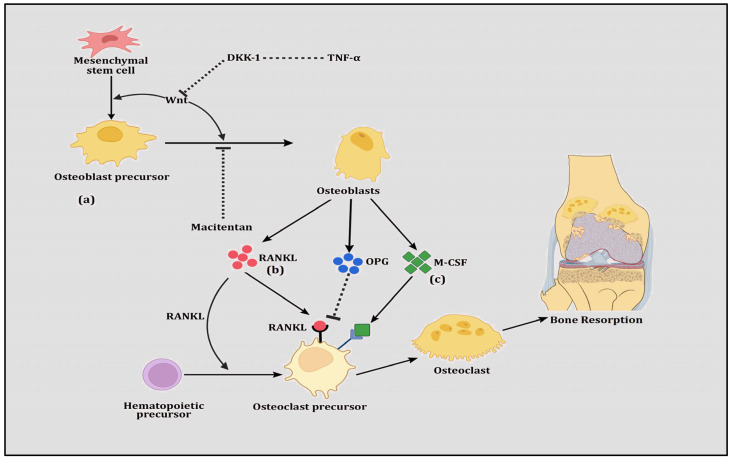
The figure shows (a) Wnt signaling in which mesenchymal stem cells are differentiated into osteoblast precursor cells and transformed to osteoblasts. Prompted by Wnt proteins, these transformations are induced however by the inhibition of TNFα-directed DKK-1 and the drug Macitentan. (b) RANK-RANKL-OPG signaling in which hematopoietic stem cells are differentiated into osteoclast precursor cells, transforming them into osteoclasts. This transformation is stimulated by RANKL but inhibited by OPG, both of which are secreted by osteoblasts. (c) M-CSF signaling in which osteoclast precursor cells are transformed into osteoclasts and this transformation is augmented by M-CSF, which is secreted by osteoblasts.

**Figure 2 diseases-12-00252-f002:**
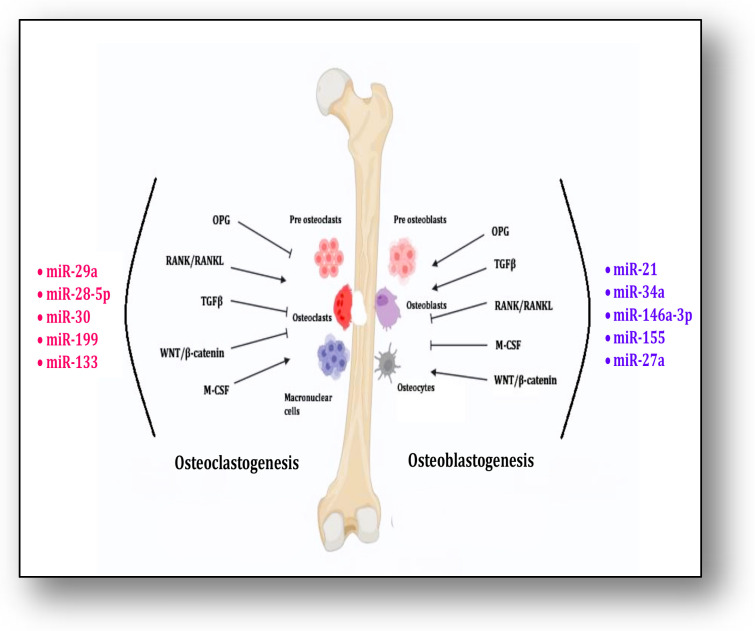
The figure shows the process of osteoclastogenesis and osteoblast differentiation by miRNAs through interaction of different signaling pathways.

**Figure 3 diseases-12-00252-f003:**
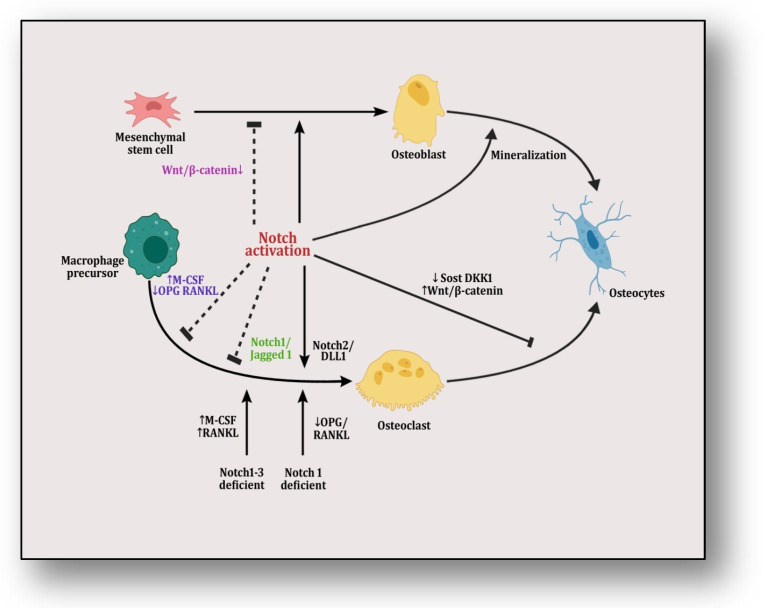
The figure illustrates Notch/Jagged signaling in which mesenchymal stem cells are differentiated into osteoblasts and are transformed into osteocytes through mineralization. Notch activation plays a dual role in this transformation by inducing it but also halting it by targeting Wnt/β-catenin pathway. On the other hand, macrophage precursors are differentiated into osteoclasts that are transformed into osteocytes.Here, notch activation also plays a dual role by inciting it either through its Notch2/Dll1 complex or by providing Notch1/Notch1–3-deficient conditions in precursor cells, which affect RANKL/OPG and M-CSF signaling. However, it is inhibited by either knocking RANK or pumping M-CSF signaling or through its Notch1/Jagged1 complex. When activating Wnt/β-catenin pathway by suppressing SOST and DKK1, this transformation is inhibited.

**Figure 4 diseases-12-00252-f004:**
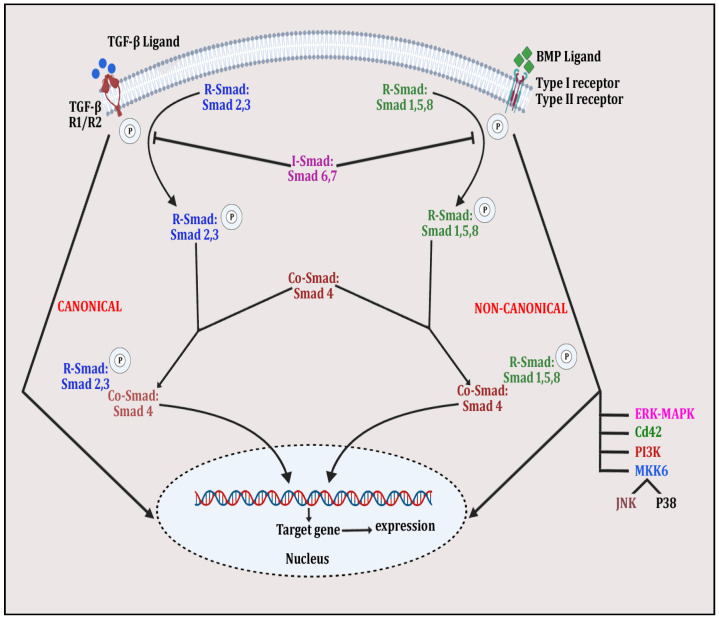
The figure depicts TGFβ/BMP signaling in whichSMAD-dependent pathway is initiated by binding of TGFβ ligand with TGFβ receptors R1 and R2 and BMP ligand with Type I/II receptor. This is followed by phosphorylation of R-Smad 2 and 3 and R-Smad 1,5 and 8 and the complexing of Co-Smad 4 with both molecules to form the activated quaternary complex, which modulates gene expression, affecting bone remodeling. Meanwhile, in the non-canonical pathway, the same ligand–receptor binding activates other signaling molecules, such as ERK-MAPK, PI3K, JNK, p38 and Cd42, by phosphorylating them. These activated molecules then alter expression of various bone-related genes.

**Table 1 diseases-12-00252-t001:** Participating microRNAs affecting different stages of bone remodeling.

Bone Remodeling Stages	Participating miRNAs	Potential Role	References
Activation	miR-185, miR-433-3p, miR-503, miR-17/20a cluster, miR-106b, miR-21, miR-214, miR-195a, miR-223, miR-7b, miR-143-3p	These micro-RNAs supress the activation of osteoclastic fate by inhibiting the transformation of osteoclast precursor cells into osteoclasts.	[43,44,45,46,47]
Resorption	miR-133a-3p and 5p, miR-193-3p, miR-155, miR-124, miR-186. miR-34a-5p, miR-125a, miR-146a-3p, miR-346-3p, miR-126-5p	These micro-RNAs repress the breakdown of bone tissue by subduing osteoclast differentiation and activity.	[48,49,50,51,52]
	miR-214-3p	This micro-RNA accentuates the bone degradation ability of osteoclasts, thereby enhancing the destruction of bone tissue.	[53]
Reversal	miR-199b, miR-1274a, miR-335-5p, miR-200c, miR-210-5p, miR-21, miR-26b, miR-34a-5p, miR-20a	These micro-RNAs prompt the migration and differentiation of osteoblast precursors along with the removal of osteoclasts from the bone that is being remodelled.	[52,54,55,56,57]
	miR-125b, miR-637, miR-9, miR-155, miR-30a, miR-199b-5p, miR-195-5p, miR-133, miR-141, miR-200a, miR-140, miR-20b, miR-345, miR-146, miR-133	These micro-RNAs hamper the transformation of osteoblast precursors into osteoblasts, which in turn halt osteogenic development.	[58,59,60,61,62]
Formation	miR-29a, miR-30b, miR-34a, miR-146a, miR-221-3p, miR-335-5p, miR-210-5p, miR-218, miR-497~195 cluster	These micro-RNAs augment the proliferation and differentiation of osteoblasts, which lead to the building of new bone tissue.	[47,63,64,65]
	miR-34c, miR-34a, miR-141, miR-22,miR-211, miR-135a-5p, miR-204-5p, miR-221-3p, miR-342-5p, miR-98-3p, miR-30, miR-542-3p	These micro-RNAs restrict osteocyte formation, which blocks the genesis of new bone.	[66,67,68,69,70]
Quiescence	miR-29a. miR-29c, miR-221-3p, miR-433-3p	These micro-RNAs hinder the process of matrix maturation and mineralization for maintaining bone architecture.	[64,71,72]
	miR-145	This micro-RNA promotes the restoration of the bone microstructure, which helps in sustaining the bone network.	[73]

**Table 2 diseases-12-00252-t002:** Potential role of knownmicroRNAs targeting RANK-OPG-RANKL pathway.

miRNAs	Target	Main Outcome	References
miR-27a	SQSTM1	Effective silencing of SQSTM1-dependent bone resorption ameliorated osteoporosis by improving bone homeostasis.	[81]
miR-21	PTEN	Successful knockdown of PTEN expression inhibited osteocyte apoptosis, thereby decreasing osteoclast activity.	[82]
miR-31	RhoA	Inhibition of RhoA gene expression increased osteoclast formation, which further stimulated bone resorption.	[86]
miR-503	RANK	Suppression of RANK signaling pathway halted osteoclastogenesis, which reduced arthritis symptoms.	[45]
miR-503	LEPR	Reduced LEPR expression enhanced osteoclast activation, resulting in increased bone resorption.	[87]
miR-21	PTEN	Negated regulation of PTEN expression subdued RANK-induced osteoclastogenesis.	[83]
miR-106b	PTEN	Decreased PTEN expression instigated inflammatory bone destruction caused by periprostheticosteolysis.	[88]
miR-133a-3p, 5P	MITF	Repression of MITF expression improved bone remodeling by decreasing osteoclast differentiation.	[48]
miR-193-3p	NFATC1	Diminished NFACT1 gene expression initiated osteoclast differentiation and gave rise to bone resorption in BMMs.	[49]
miR-155	PU1 MITF	Remarkable silencing of PU1 and MITF expression directly reduced RANKL-controlled osteoclastogenic differentiation.	[50]
miR-21	PDCD4	Effective knockdown of PDCD4 gene expression amplified RANKL-dependent osteoclast formation and activity.	[84]
miR-21	FasL Caspase3	Effective restriction of expression of FasL and Caspase3 genes inhibited estrogen-mediated osteoclastogenesis.	[85]
miR-503	RANK	Downregulation of RANK expression resulted in the inhibition of NF-kB mediated osteoclast differentiation.	[89]
miR-214-3p	TRAF3	Attenuation of TRAF3 gene expression accentuated osteolytic bone metastasis of breast cancer by increasing osteoclast activity.	[53]
miR-214	PTEN	Effective knockdown of PTEN exhibited increased differentiation of osteoclasts, which led to decline in bone density.	[90]
miR-182	FOXO3 MAML1	Downregulation of FOXO3 and MAML1 expression heightened TNF-α-mediated osteoclastogenesis.	[91]
miR-182	PKR	Decreased IFN-β-dependent PKR gene expression enhanced osteogenesis, which worsened RA.	[64,92]
miR-135b	PDCD4	Restriction of PDCD4 gene expression subdued apoptosis, which alleviated ONFH.	[46]
miR-17/20a cluster	RANKL	Negative regulation of RANKL gene expression caused repression of Dex-induced osteoclastogenesis.	[93]
miR-106b	RANKL	Suppression of RANKL expression attenuated osteolysis and osteoclastogenesis.	[94]
miR-124	NFATC1	Negated modulation of NFATc1 expression repressed TNF-α/IL-6-induced osteoclastogenesis.	[51]
miR-34a-5p	TGIF2	Negative regulation of TGIF-oriented NF-kB signalosome prevented bone loss and preserved bone architecture, which alleviated osteoporosis.	[52]
miR-186	CTSK	Repression of CTSK expression had a protective effect on bone and restored bone mass.	[95]

**miR**: Micro-RNA, **SQSTM1**: Sequestosome 1, **PTEN**: Phosphatase and TENsin homolog deleted on chromosome 10, **RhoA**: Ras homolog gene family member A, **RANK**: Receptor activator of nuclear factor kappa-beta, **LEPR**: Leptin receptor, **MITF**: Microphthalmia-associated transcription factor, **NFATC1**: Nuclear factor of activated T Cells 1, **BMMs**: Bone marrow-derived macrophages/monocytes, **PU1**: Lineage-determining ETS-domain transcription factor 1, **RANKL**: Receptor activator of nuclear factor kappa-beta ligand, **PDCD4**: Programmed cell death 4, **FasL**: FS-7-associated surface antigen ligand, **Caspase3**: Cysteine-dependent aspartate-directed proteases 3, **NF-kB**: Nuclear factor kappa-beta, **TRAF3**: Tumor necrosis factor receptor (TNFR)-associated factor 3, **TGIF2**: TGFβ-induced transcription factor 2, **FOXO3**: Factor forkhead box O-3, **MAML1**: Mastermind-like transcriptional coactivator 1, **TNF-α**: Tumor necrosis factor-alpha, **PKR**: Double-stranded RNA-dependent protein kinase, **IFN-β**: Interferon-beta, **RA**: Rheumatoid arthritis, **ONFH**: Osteonecrosis of the femoral head, **Dex**: Dexamethasone, **IL-6**: Interleukin-6, **CTSK**: Cathepsin K.

**Table 3 diseases-12-00252-t003:** Potential role of knownmicroRNAs targeting M-CSF pathway.

miRNAs	Target	Main Outcome	References
miR-7b	DC-STAMP	Repression of DC-STAMP expression hindered osteoclastogenic differentiation of precursor cells.	[47]
miR-214	PTEN	Negated modulation of PTEN expression reduced bone density as a result of enhanced osteoclast action.	[90]
miR-155	SOCS1 MITF	Attenuation of SOCS1 and MITF gene expression resulted in impeded osteoclast differentiation and activity.	[101]
miR-155	LEPR	Suppression of LEPR expression increased osteoclast activation, which prompted bone resorption.	[87]
miR-21	STAT3	Downregulation of STAT3-dependent M-CSF signaling ameliorated arthritic condition.	[102]
miR-143-3p	M-CSF	Reduced expression of M-CSF attenuated osteoclast differentiation, which mitigated arthritis symptoms.	[103]
miR-125a	TRAF6	Effective silencing of TRAF6 gene expression attenuated osteoclastogenesis.	[104]
miR-146a-3p	TRAF6	Inhibition of TRAF6 hampered osteoclast differentiation, which further retarded bone resorption.	[48]
miR-195a	CSF1	Successful knockdown of CSF1 expression reduced osteoclast differentiation and thus hindered bone resorption.	[105]
miR-346-3p	TRAF3	Negative regulation of TRAF3 gene expression accentuated osteoclastogenesis in precursor cells.	[106]
miR-223	M-CSFR	Repression of M-CSFR expression inhibited osteoclast formation and alleviated osteoporotic condition.	[107]

**miR**: Micro-RNA, **DC-STAMP**: Dendrocyte-expressed seven transmembrane protein, **PTEN**: Phosphatase and TENsin homolog deleted on chromosome 10, **SOCS1**: Suppressor of cytokine signaling 1, **MITF**: Microphthalmia-associated transcription factor, **LEPR**: Leptin receptor, **STAT3**: Signal transducers and activators of transcription 3, **M-CSF**: Macrophage colony-stimulating factor, **TRAF6**: Tumor necrosis factor receptor (TNFR)-associated factor 6, **CSF-1**: Macrophagecolony-stimulating factor-1, **TRAF3**: Tumor necrosis factor receptor (TNFR)-associated factor 3, **M-CSFR**: Macrophage colony-stimulating factor-1 receptor.

**Table 4 diseases-12-00252-t004:** Potential role of knownmicroRNAs targeting Wnt/β-catenin pathway.

miRNAs	Target	Main Outcome	References
miR-29a miR-29c	ON	Restricted expression of ON increased matrix maturation and mineralization of bone tissue along with enhanced osteoblast differentiation.	[71]
miR-199b	WNT2	Successful knockdown of WNT2 expression induced osteogenesis.	[54]
miR-1274a	BMPR1B	Efficient silencing of BMPR1B gene stimulated osteoblast differentiation.	[54]
miR-30b	RUNX2	Repression of RUNX2 expression accentuated osteogenic development.	[54]
miR-29a	WNT3A GSK3β β-catenin	Downregulation of various components of WNT/β-catenin signaling prompted osteogenic differentiation and hampered bone resorption.	[120]
miR-34a miR-146a	β-catenin	Negative modulation of β-catenin expression hamperedboth osteoclastogenesis and angiogenesis in the reparative phase of ONFH.	[121]
miR-29a	DKK1 GSK3β	Negated regulation of WNT antagonists attenuated TNF-α-induced bone loss by increasing osteogenic differentiation.	[122]
miR-141 miR-22	β-catenin	Osteogenic development was obstructed as a result of blockage of β-catenin expression.	[66]
miR-221-3p	DKK2	Osteoblast formation and maturation was hindered due to the decrease in expression of DKK2 gene.	[64]
miR-29a	GSK3β	Effective silencing of GSK3β gene led to the suppression of bone formation in AS patients.	[123]
miR-355-5p	DKK1	Restriction of DKK1 gene expression accentuated bone formation in mouse embryos.	[124]
miR-210-5p	ACVR1B	Attenuation of ACVR1B expression prompted osteoblastic development, which ameliorates Cushing’s syndrome caused by hypercortisolism.	[68]
miR-211 miR-135a-5p miR-204-5p	RUNX2	Termination of WNT-dependent osteogenic development due to suppressed RUNX2-mediated WNT signaling in response to endogenous hypercortisolism.	[68]
miR-199a-5p	DKK1 WNT10B	Repression of DKK1 and WNT10B expression promoted mesenchymal stem cell commitment towards chondrogenic development.	[125]
miR-185	PTH	Fracture healing was halted as a result of suppressed PTH expression followed by attenuated osteoblast proliferation.	[43]
miR-335-5p	DKK1	Blockage of DKK1 gene expression prompted osteogenic differentiation of stem cells.	[126]
miR-26a-5p	WNT5A CaMKII	Negative regulation of WNT5A and CaMKII led to the repressed bone formation.	[127]
miR-200c	SOX2 KLF4	Inhibition of SOX2-dependent WNT signaling induced osteogenic differentiation.	[55]
miR-146a	WNT1 WNT5A	Efficient knockdown of WNT1 and WNT5A genes stimulated age-related bone loss.	[63]
miR-29a	DKK1	Negated modulation of DKK1 reversed ovariectomized-related bone loss in mice.	[128]
miR-26b	β-catenin	Obstructed β-catenin expression resulted in the induction of osteogenic differentiation in subchondral bone stem cells.	[56]
miR-433-3p	DKK1	Attenuation of DKK1-mediated Wnt/β-catenin pathway promoted bone homeostasis and mitigated osteoporosis.	[44]
miR-335-5p	DKK1	Remarkable increase in osteogenesis as a result of inhibition of high-glucose-induced apoptosis of osteoblasts.	[129]
miR-218	DKK2 SOST SFRP2	Effective knockdown of WNTinhibitors amplified osteoblast activity and osteomimicry of cancer cells.	[130]
miR-21	SOX2	Suppression of SOX2 expression augmented osteogenesis in fetal MSCs and adult BMSCs.	[131]
miR-125b	OSX	Downregulation of OSX expression reduced proliferation and osteogenic differentiation of BMSCs.	[59]
miR-637	OSX	Repression of OSX expression decreased osteogenic differentiation and enhanced adipogenic differentiation.	[60]

**miR**: Micro-RNA, **ON**: Osteonectin, **WNT2**: Wingless-related integration site family member 2, **BMPR1B**: Bone morphogenetic protein receptor type 1B, **RUNX2**: Runt-related transcription factor 2, **WNT3A**: Wingless-related integration site family member 3A, **GSK3β**: Glycogen synthase kinase 3 beta, **β-catenin**: Beta-catenin, **WNT**: Wingless-related integration site gene family, **ONFH**: Osteonecrosis of the femoral head, **DDK1**: Dickkopf-1, **TNF-α**: Tumor necrosis factor-alpha, **DDK2**: Dickkopf-2, **AS**: Ankylosing spondylitis, **ACVR1B**: Activin A receptor type 1B, **WNT10B**: Wingless-related integration site family member 10B, **PTH**: Parathyroid hormone, **WNT5A**: Wingless-related integration site family member 5A, **CaMKII**: Calcium/calmodulin-dependent protein kinase type II, **SOX2**: Sex-determining region Y chromosome-related high-mobility-group BOX gene 2, **KLF4**: Kruppel-like factor 4, **WNT1**: Wingless-related integration site family member 1, **WNT5A**: Wingless-related integration site family member 5A, **SOST**: Sclerostin, **SFRP2**: Secreted frizzled related protein 2, **MSCs**: Mesenchymal stem cells, **BMSCs**: Bone marrow-derived mesenchymal stem cells/Bone marrow-derived stromal cells, **OSX**: Osterix.

**Table 5 diseases-12-00252-t005:** Potential role of known microRNAs targeting Notch/Jagged pathway.

miRNAs	Target	Main Outcome	References
miR-9	NOTCH1	Negative modulation of Notch/Jagged signaling transformed bone MSCs into neurons.	[142]
miR-34c	NOTCH1 NOTCH2 JAG1	Suppression of different Notch pathway components hindered osteoblastogenesis and differentiation of osteoblasts.	[70]
miR-34a	JAG1	Inhibition of JAG1 resulted in reduced osteoblast differentiation.	[143]
miR-155	kB-RAS1	Efficient silencing of kB-RAS1 gene expression permuted hematopoietic homeostasis, which developed myeloproliferative conditions.	[144]
miR-34a-5p	JAG1	Suppression of JAG1 expression stimulated osteogenic differentiation of BMSCs.	[145]
miR-30a	DLL4	Downregulation of DLL4 gene expression elevated chondrogenic differentiation.	[146]
miR-497~195 cluster	F-BOX FBXW7 P4HTM	Repression of the respective genes elevated Notch signaling which further prompted coupled angiogenic–osteogenic activity.	[147]
miR-146a	NOTCH1	Effective knockdown of Notch signaling ameliorated PTOA as a result of chondrogenic differentiation.	[65]
miR-487b-3p	NRARP	Decreased NRARP gene expression halted Notch-dependent osteogenic differentiation.	[148]
miR-17-5p miR-15a-5p miR-181b-5p miR-106a-5p miR-106b-5p miR-16-5p miR-93-5p	Various genes including that of Notch/Jagged signalling	Differential expression of various genes directly influenced AIS pathogenesis and secondary osteopenia.	[149]
miR-34a-5p	NOTCH1	Efficient silencing of NOTCH1 gene expression led to initiated apoptosis, which exacerbated IVDD.	[150]
miR-342-5p	BMP7 COL4A6	Suppression of BMP7 and COL4A6 expression inhibited proliferation, migration and differentiation of osteoblasts, which further impeded fracture healing.	[151]
miR-199b-5p	JAG1	Attenuated JAG1 gene expression invigorated chondrogenic differentiation of hBMSCs.	[58]
miR-210	VEGF NOTCH1	Upregulation of VEGF and NOTCH1 expressions mitigated osteoporotic conditions.	[152]
miR-28-5p	NOTCH1	Negated modulation of NOTCH1 gene expression augmented recovery from SCI.	[153]
miR-199b-5p	JAG1	Repression of JAG1 expression impaired osteogenesis in ligamentum flavum cells.	[154]
miR-195-5p	DLL1	Restriction of DLL1 gene expression incited apoptosis and hindered cell growth.	[155]
miR-98-3p	JAG1	Poor osteogenic differentiation was noted as a result of silencing of Notch/Jagged signaling.	[69]

**miR**: Micro-RNA, **NOTCH1:** Neurogenic locus notch homolog protein1, **MSCs**: Mesenchymal stem cells, **NOTCH2**: Neurogenic locus notch homolog protein 2, **JAG1**: Jagged canonical Notch ligand 1, **Kb-RAS1**: Nuclear factor kappa-beta-rat sarcoma 1, **BMSCs**: Bone-marrow-derived mesenchymal stem cells/Bone-marrow-derived stromal cells, **DLL4**: Delta-like 4, **F-BOX**: Substrate recognition component of Skp1-Rbx1-Cul1-F-Box ubiquitin ligases, **FBWX7**: F-box and WD-repeat domain containing 7, **P4HTM**: Prolyl 4-hydroxylase transmembrane, **NOTCH**: Neurogenic locus Notch homolog, **PTOA**: Post-traumatic osteoarthritis, **NRARP**: Notch regulated ankyrin repeat protein 1, **AIS**: Adolescent idiopathic scoliosis, **IVDD**: Intervertebral disc degeneration, **BMP7**: Bone morphogenetic protein 7, **COL4A6**: Collagen type IV alpha-6 chain, **hBMSCs**: Human bone-marrow-derived mesenchymal stem cells, **VEGF**: Vascular endothelial growth factor, **SCI**: Spinal cord injury, **DLL1**: Delta-like 1.

**Table 6 diseases-12-00252-t006:** Potential role of known microRNAs targeting TGFβ/BMP pathway.

miRNAs	Target	Main Outcome	References
miR-133	RUNX2 SMAD5	Attenuation of RUNX2 expression inhibited osteoprogenitor differentiation.	[62]
miR-30	RUNX2 SMAD1	Blockade of osteogenic differentiation by targeting RUNX2 and SMAD1 gene expression.	[160]
miR-322	TOB2	Negative modulation of TOB2 expression increased osterix and BMP2 levels, which resulted in increased osteogenic development.	[161]
miR-141 miR-200a	DLX5	Effective silencing of DLX5 gene expression halted BMP-induced pre-osteoblast development.	[162]
miR-542-3p	BMP7	Remarkable knockdown of BMP7 expression hindered bone formation by decreasing osteoblast differentiation.	[67]
miR-20a	PPARγ CRIM1 BAMBI	Negative regulation of these respective genes promoted osteogenic development of hMSCs by upregulating BMP signaling.	[57]
miR-140	DNPEP	Impeded DNPEP expression accelerated BMP signaling, which further hampered endochondral bone formation.	[163]
miR-146a	SMAD4	Inhibition of SMAD4 gene expression induced chondrocyte apoptosis, which further exacerbated OA.	[164]
miR-199a	SMAD1	Retardation of early chondrocyte differentiation as a result of negated modulation of SMAD1 expression.	[165]
miR-20b miR-345 miR-146	SMAD3 SMAD4 SMAD6	Downregulation of these SMAD proteins led to the blockage of chondrogenic development, which aggravated OA.	[61,166,167]
miR-146a	SMAD4	Repression of SMAD4 expression hindered osteogenic differentiation of ADSCs.	[168]
miR-433-3p	CREB1 HSD11B1 RSPO3	Suppression of these respective target genes led to the adverse regulation of bone formation.	[72]
miR-21 miR-29 miR-221 Let-7a	PTEN AKT ECM-related genes	Osteoporotic condition was ameliorated by positive modulation of the expression of these respective genes.	[169]
miR-20a	BAMBI SMAD6	Decreased BAMBI and SMAD6 expression stimulated BMP2-dependent osteogenic differentiation.	[170]
miR-133	RUNX2	Restriction of RUNX2-mediated BMP signaling hampered bone formation of osteoblast lineage cells.	[62]
miR-135	SMAD5	Reduced expression of SMAD5 gene expression led to decrease in osteogenesis.	[62]
miR-126-5p	MMP-13	Reduction in MMP-13-dependent TGFβ signaling led to decreased osteoclast activity and inhibition of osteolysis.	[171]
miR-203 miR-135	RUNX2	Repression of RUNX2 expression incited anti-tumor and anti-metastatic effects, which prompted bone revival.	[172]
miR-145	SMAD3	Remarkable silencing of SMAD3-dependent TGFβ-mediated osteoclast differentiation resulted in improved bone mass and restored bone microstructure.	[73]

**miR**: Micro-RNA, **RUNX2**: Runt-related transcription factor 2, **SMAD5**: Suppressor of mothers against decapentaplegic homolog 5, **SMAD1**: Suppressor of mothers against decapentaplegic homolog 1, **TOB2**: Transducer of v-erb-b2 avian erythroblastic leukemia viral oncogene homolog 2, 2, **BMP2**: Bone morphogenetic protein 2, **DLX5:** Distal-less homeobox 5, **BMP**: Bone morphogenetic protein, **BMP7**: Bone morphogenetic protein 7, **PPARγ**: Peroxisome proliferator-activated receptor gamma, **CRIM1**: Cysteine-rich motor neuron 1, **BAMBI**: BMP and activin membrane bound inhibitor, **hBMSCs**: Human bone-marrow-derived mesenchymal stem cells, **DNPEP**: Aspartyl aminopeptidase, **SMAD4**: Suppressor of mothers against decapentaplegic homolog 4, **OA**: Osteoarthritis, **SMAD3**: Suppressor of mothers against decapentaplegic homolog 3, **SMAD6**: Suppressor of mothers against decapentaplegic homolog 6, **SMAD**: Suppressor of mothers against decapentaplegic, **ADSCs**: Adipose tissue-derived stem cells, **CREB1**: cAMP response element binding protein 1, **HSD11B1**: Hydroxysteriod 11-beta dehydrogenase 1, **RSPO3**: R-Spondin3, **PTEN**: Phosphatase and TENsin homolog deleted on chromosome 10, **AKT**: Protein kinase B or AK mouse plus transforming or thymoma, **ECM**: Extracellular matrix, **MMP-13**: Matrix metalloproteinase-13, **TGFβ**: Transforming growth factor beta.

## Data Availability

Data is available with the corresponding author.

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
