# Peer review of "Molecular Signaling Pathways and MicroRNAs in Bone Remodeling: A Narrative Review"

_diseases, 2024, doi:10.3390/diseases12100252_

Round 1
Reviewer 1 Report
Comments and Suggestions for Authors
This article aims to provide an updated overview of the role of miRNAs in molecular signaling pathways that regulate bone remodeling. However, this topic has been recently reviewed in detail by Jiménez-Ortega et al. (Biology, MDPI, 2024 Jul; PMID: 39056698) with a better presentation format. Jiménez-Ortega et al. also discussed the involvement of miRNAs in various signaling pathways, including RANKL/RANK, MSCF, Wnt, Notch, and BMP/TGFb.
It is promising that Singh et al. summarize additional miRNAs not listed in previous reviews (PMID: 39056698, PMID: 31667459, PMID: 26208037, PMID: 29848766). The authors described the findings from these references. However, it is unclear how some of the targets relate to the signaling pathways in the tables (e.g., LEPR, PTEN in RANKL/RANK and MCSF signaling).
Other comments:
- Line 177, 178: It states, “RANKL expressed by osteoclasts and dendritic cells is a signaling receptor for RANK.” The words RANKL and RANK should be switched.
Author Response
Reviewer 1
Comment1: This article aims to provide an updated overview of the role of miRNAs in molecular signaling pathways that regulate bone remodeling. However, this topic has been recently reviewed in detail by Jiménez-Ortega et al. (Biology, MDPI, 2024 Jul; PMID: 39056698) with a better presentation format. Jiménez-Ortega et al. also discussed the involvement of miRNAs in various signaling pathways, including RANKL/RANK, MSCF, Wnt, Notch, and BMP/TGFb.
Response 1: Yes, we also believe that the paper by Jimenez-Ortiga is better and informative, but, we didn’t see this paper till the submission of this present paper and that is the reason that this paper is not cited in our paper. Now we have included this paper in our reference list.
Comment 2: It is promising that Singh et al. summarize additional miRNAs not listed in previous reviews (PMID: 39056698, PMID: 31667459, PMID: 26208037, PMID: 29848766). The authors described the findings from these references. However, it is unclear how some of the targets relate to the signaling pathways in the tables (e.g., LEPR, PTEN in RANKL/RANK and MCSF signaling).
Response 2: Sir, for every pathway, we have given the explanation of the relationship of respective MicroRNA and that particular pathway (before table) and then we have given the main outcome of that particular study in the table. In this way every study given in the table shows how, the target is associated with the signaling pathway (explained in the text).
For instance, in section ‘2.1 RANK-OPG-RANKL pathway’ it has been written in lines 133-136, ‘It has been observed that miR27a plays role in osteoclastogenesis rather than osteoblastogenesis, hence, involved in bone resorption mediated osteoporosis [32]. miR27a participates in osteoclast differentiation through the mediation of sequestosome 1 (Sqstm1/p62), which has been implicated in RANK mediated osteoclastogenesis in Paget’s disease’Now this study [32] is written in the table with its main outcome explaining that this target SQSTM1 is influenced by MiR-27a through RANK-OPG-RANKL pathway (which is the main section).
Other comments:
Comment 3: Line 177, 178: It states, “RANKL expressed by osteoclasts and dendritic cells is a signaling receptor for RANK.” The words RANKL and RANK should be switched.
Response 3: We are sorry for this inadvertent mistake. It has been corrected now.
Reviewer 2 Report
Comments and Suggestions for Authors
Precise description of the target pathways could be improved but are adequate.
The tables summarizing the miRNAs and there targets are quite useful.
The narrative is relative dense but does provide additional context to the tables.
There is some concern relative to syntax, in particular, that could be improved.
Overall, the manuscript meets the goal of summarizing current state of knowledge regarding miRs in bone remodeling.
Comments on the Quality of English LanguageSyntax could be improved.
Author Response
Comment 1: Precise description of the target pathways could be improved but are adequate.
Response 1: Sir, we have kept it precise and crisp but within the planned space, otherwise, the length of this paper would have increased manifold.
Comment 2: The tables summarizing the miRNAs and their targets are quite useful.
Response 2: Thanks Sir
Comment 3: The narrative is relative dense but does provide additional context to the tables.
Response 3: Thanks for the understanding.
Comment 4: There is some concern relative to syntax, in particular, that could be improved.
Response 4: Sir, it would have been better, if, you have mentioned specific lines or section, but we have improved the syntax with the help of an English language expert.
Comment 5: Overall, the manuscript meets the goal of summarizing current state of knowledge regarding miRs in bone remodeling.
Response 5: Thanks for the agreement, Sir.
Comment 6: Comments on the Quality of English Language: Syntax could be improved.
Response 6: We have improved the syntax with the help of an English language expert.
Reviewer 3 Report
Comments and Suggestions for Authors
Monica Singh and Puneetpal Singh et al. submitted an interesting review about the mechanisms associated with bone remodeling. The topic was less frequently discussed yet with much importance. Overall, the topic fell within the scope of Diseases. The reviewer suggested a Minor Revision for it. Detailed comments:
1. Please avoid using too many abbreviations in the Keywords (like M-CSF and BMP), as some audience might be unfamiliar with them.
2. It was recommended to introduce some basics on the bone diseases in the first paragraph of Introduction.
3. In Section 2, the authors just listing the different pathways. It was suggested to make some personal comments and comparison about these pathways.
4. In Section 3, please add some discussion about the drug discovery aspects. For example, for RANK-OPG-RANKL, M-CSF, Wnt/β-catenin, Notch and TGF-β/BMP pathways, how could we design and develop new drugs?
Author Response
Comment 1: Please avoid using too many abbreviations in the Keywords (like M-CSF and BMP), as some audience might be unfamiliar with them.
Response 1: We agree to this Sir, these keywords have been corrected.
Comment 2: It was recommended to introduce some basics on the bone diseases in the first paragraph of Introduction.
Response 2: We have written the basics on the bone diseases in the first paragraph of the Introduction.
Comment 3: In Section 2, the authors just listing the different pathways. It was suggested to make some personal comments and comparison about these pathways.
Response 3: We have incorporated personal comments for these pathways in the end of the section 2, which is now section 4.
Comment 4: In Section 3, please add some discussion about the drug discovery aspects. For example, for RANK-OPG-RANKL, M-CSF, Wnt/β-catenin, Notch and TGF-β/BMP pathways, how could we design and develop new drugs?
Response 4: We have incorporated a paragraph on this issue in section 3, which is section 5 now.
Reviewer 4 Report
Comments and Suggestions for Authors
1. Biological functions of miRNA should be briefly summarized in the introduction section.
2. A schematic summary of miRNAs implicated in osteoblast differentiation can be included for better understanding the manuscript.
3. There are many miRNA that inhibit osteogenesis. The possible mechanism behind their action could also be described.
4. The biogenesis of miRNA should be summarized at the beginning of section 2.
5. The limitations of the study are advised to be included.
6. Many changes are required, mostly for sentence reformulation in a more appropriate language.
7. There are several articles on the same topic. What is the novelty of your work compared to the others.
8. Different stages of bone remodeling and particular miRNA affecting them should be provided in a table form.
9. Clinical implications of miRNA should be expanded with their references.
10. Moreover, the authors fail to justify why miRNA should be better than current allopathic therapy for same purpose.
Comments on the Quality of English LanguageThe syntax part should focused.
Author Response
Comment 1: Biological functions of miRNA should be briefly summarized in the introduction section.
Response 1: We have summarized the biological functions of miRNA in the introduction section.
Comment 2: A schematic summary of miRNAs implicated in osteoblast differentiation can be included for better understanding the manuscript.
Response 2: We have included a schematic summary of miRNAs in osteoblast as well as osteoclast differentiation in the form of a figure 2. Its explanation has also been incorporated in the text.
Comment 3: There are many miRNA that inhibit osteogenesis. The possible mechanism behind their action could also be described.
Response 3: We have incorporated the separate section 3 on ‘MicroRNAs inhibiting osteogenesis’
Comment 4: The biogenesis of miRNA should be summarized at the beginning of section 2.
Response 4: We have summarized the biogenesis of miRNA as separate section as ‘2. MicroRNA biogenesis’.
Comment 5: The limitations of the study are advised to be included.
Response 5: Sir, we have added limitations of the study as a separate paragraph.
Comment 6: Many changes are required, mostly for sentence reformulation in a more appropriate language.
Response 6: Sir, it would have been better, if, you have mentioned specific lines or section, but we have improved the syntax with the help of an English language expert.
Comment 7: There are several articles on the same topic. What is the novelty of your work compared to the others.
Response 7: There are several articles on this topic of bone remodeling, but they are similar not the same, as some have missed some key signaling pathways or others failed to enlist all miRNAs involved sofar. Detailed account of modus operandi of five major molecular signaling pathways along with functions of potential miRNAs, which participate in these pathways, is the only difference in comparison to other studies.
Comment 8: Different stages of bone remodeling and particular miRNA affecting them should be provided in a table form.
Response 8: It has been added as table 1. in the manuscript.
Comment 9: Clinical implications of miRNA should be expanded with their references.
Response 9: We have written clinical implications of miRNA with references in the section ‘Practical implications and future directions’ which is now changed to ‘Clinical implications and future directions’.
Comment 10: Moreover, the authors fail to justify why miRNA should be better than current allopathic therapy for same purpose.
Response 10: Sir, this write-up never claimed that miRNA induced post transcriptional silencing of the targets is better than allopathic therapy, rather we have suggested that in this point of time when 178 million new cases of fractures, 455 million acute or chronic fractures along with 25.8 millions of YLD are reported during the years 1990 to 2019, searching and researching for novel and alternative therapies could be a better option for silencing targets even before they express. In the text of this paper we have written
‘Commonly used strategies including pharmacotherapy, physical therapy, orthopedic management, life style modifications, nutritional interventions, clinical therapies, pain management and surgical procedures have partially resolved the associated inflammation, damaged tissue and bone loss[4–6]. Therapeutic approaches including mesenchymal stem cell replacement and pulsed electromagnetic field have also contributed in reducing the effect of clinical variables which exacerbate bone pathologies[7,8]. In the face of evolving challenges of bone disorders, developing improved methodologies to curb and cure are direly needed’
Round 2
Reviewer 1 Report
Comments and Suggestions for Authors
The review article is improved after revision.